# UltrasonicGS: A Highly Robust Gesture and Sign Language Recognition Method Based on Ultrasonic Signals

**DOI:** 10.3390/s23041790

**Published:** 2023-02-05

**Authors:** Yuejiao Wang, Zhanjun Hao, Xiaochao Dang, Zhenyi Zhang, Mengqiao Li

**Affiliations:** 1College of Computer Science and Engineering, Northwest Normal University, Lanzhou 730070, China; 2Gansu Province Internet of Things Engineering Research Center, Lanzhou 730070, China

**Keywords:** ultrasonic sensing, gesture recognition, sign language recognition, GAN, CTC

## Abstract

With the global spread of the novel coronavirus, avoiding human-to-human contact has become an effective way to cut off the spread of the virus. Therefore, contactless gesture recognition becomes an effective means to reduce the risk of contact infection in outbreak prevention and control. However, the recognition of everyday behavioral sign language of a certain population of deaf people presents a challenge to sensing technology. Ubiquitous acoustics offer new ideas on how to perceive everyday behavior. The advantages of a low sampling rate, slow propagation speed, and easy access to the equipment have led to the widespread use of acoustic signal-based gesture recognition sensing technology. Therefore, this paper proposed a contactless gesture and sign language behavior sensing method based on ultrasonic signals—UltrasonicGS. The method used Generative Adversarial Network (GAN)-based data augmentation techniques to expand the dataset without human intervention and improve the performance of the behavior recognition model. In addition, to solve the problem of inconsistent length and difficult alignment of input and output sequences of continuous gestures and sign language gestures, we added the Connectionist Temporal Classification (CTC) algorithm after the CRNN network. Additionally, the architecture can achieve better recognition of sign language behaviors of certain people, filling the gap of acoustic-based perception of Chinese sign language. We have conducted extensive experiments and evaluations of UltrasonicGS in a variety of real scenarios. The experimental results showed that UltrasonicGS achieved a combined recognition rate of 98.8% for 15 single gestures and an average correct recognition rate of 92.4% and 86.3% for six sets of continuous gestures and sign language gestures, respectively. As a result, our proposed method provided a low-cost and highly robust solution for avoiding human-to-human contact.

## 1. Introduction

The world has suffered from a sudden outbreak of a new coronavirus that has had a widespread impact on people’s lives. In particular, in recent times, a number of countries and regions around the world have seen a recurrence of the situation. The situation of epidemic prevention and control is still serious. In the face of this massive epidemic, the World Health Organization (WHO) states in its guidance article that avoiding human-to-human contact can effectively cut off the spread of the virus [1]. Therefore, contactless gesture recognition becomes an effective means to reduce the risk of contact infection in outbreak prevention and control. However, especially in the face of daily behavior recognition for certain populations, such as the deaf, the labor cost of hiring a sign language teacher is high. Therefore, how to correctly and efficiently recognize sign language gestures and perform human–computer interaction has become a problem that needs to be solved.

Past research work on gesture recognition was divided into three main categories: sensor-based [2], vision-based [3], and Wi-Fi-based [4,5]. In sensor-based systems, limb motion features are captured by body-worn sensors. In vision-based systems, limb motion features are captured by optical cameras. In Wi-Fi-based systems, extracting channel state information (CSI) can recognize limb motion. By collecting human behavior information, different data processing processes, and classification learning, all of the above methods can identify people’s behaviors. However, there are certain limitations to these techniques. Vision-based sensing technology is highly influenced by light and has poor privacy and high energy consumption requirements for long-term detection. Sensor-based sensing technology causes a lot of inconvenience to users because they need to wear external devices for a long time. For Wi-Fi-based sensing technology, recognition accuracy is affected because Wi-Fi signals are susceptible to interference from electromagnetic waves.

To compensate for the limitations of traditional techniques, the use of acoustic waves for human activity perception is gradually gaining attention. Due to the advantages of slow propagation speed, low sampling rate, and easy access to equipment, in recent years, relevant research based on ultrasonic signals has also made great progress in smart homes [6], location tracking [7], gesture recognition [8], and facial recognition [9]. Research work in gesture recognition includes: Gao et al. [10] captured gestures using lightweight MobileNet by using dual speakers and microphones in smartphones. LLAP [11] obtained the accurate motion direction and distance by measuring the phase change of the received signal to realize two-dimensional gesture tracking. Strata [12] achieved more accurate recognition of gestures by estimating the Channel Impulse Response (CIR) of the reflected signal. In this paper, we focus on human gesture recognition, especially extending to sign language recognition for certain groups, such as deaf people [13], and providing higher perceptual accuracy.

Due to the complexity of gesture movements, implementing acoustic-based fine-grained, and highly robust gesture and sign-language-recognition methods have two challenges. The first challenge is insufficient training data. The approach in this paper involves three tasks: single gesture recognition, continuous gesture recognition, and sign language gesture recognition. It takes time and effort to collect sufficient data for each task. Past work either did not use data augmentation methods or used traditional data augmentation methods based on geometric transformations and image manipulation. Although it can alleviate the problem of neural network overfitting and improve the generalization ability to a certain extent, the method used lacks flexibility and covers more limited situations. The second challenge is to solve the problem of inconsistent length and difficult alignment of input and output sequences of continuous and sign language gestures. Because most of the previous perception-based research work [14] can only recognize a single gesture, or several consecutive individual actions, especially since there is no research work using acoustic perception for Chinese sign language recognition. Continuous gesture and sign language recognition is an indeterminate length sequence prediction problem. Traditional sequence prediction networks usually only produce fixed-length outputs and can not determine the length of the prediction sequence adaptively.

For this purpose, a highly robust gesture and sign language recognition method based on ultrasonic signals are proposed in this paper. First, we use the ultrasonic device Acoustic Software Defined Radios Platform (ASDP) to capture the gesture movement data and the amplitude information is used as the feature value for denoising and smoothing. Then we use short-time Fourier transform (STFT) to extract the Doppler shift of the movement data. To address the challenge of insufficient training data, we use GAN to automatically generate data. Then ResNet34 is used to extract the feature values and the bi-directional long short-term memory (Bi-LSTM) algorithm is used to classify the single gesture. For continuous gestures and sign language gestures, the CTC algorithm is added after the Bi-LSTM network. We use the dynamic programming method to find the output result with the highest probability as the final output result of the model. The main contributions of this paper are as follows:1.We propose a data augmentation method based on GAN. Due to the randomness of GAN itself, it makes the generated samples more diverse and can cover more real situations, while it can reduce the classification model error and improve the performance of the model.2.We feed the multi-scale semantic features extracted by the residual neural network into the Bi-LSTM algorithm. The algorithm enables the classification network to fuse the information of feature dimension and temporal dimension to achieve high-precision gesture recognition. Meanwhile, in order to fill the gap of acoustic perception recognition of continuous gestures and Chinese sign language gestures and solve the problem of inconsistent length and difficult alignment of continuous gesture and sign language gesture input and output sequences, we add the CTC algorithm after the Bi-LSTM network. It enables the model to achieve good results for continuous gesture recognition and sign-language-recognition problems as well.3.In this paper, we obtain real data on gestures from multiple groups of volunteers and form an open-source database. Through two real scene tests, it is verified that the proposed method has high robustness, the accuracy of single gesture recognition reaches 98.8%, and the recognition distance is 0.5 m. At the same time, the sign language data collected can provide data support for education professionals to study the daily interaction behavior of certain groups, such as the deaf.

The remaining sections of this paper are organized as follows. Section 2 summarizes the existing work related to gesture and sign language recognition. Section 3 explains the implementation process of the UltrasonicGS method. In Section 4, we experiment and evaluate the performance of the UltrasonicGS method. Finally, Section 5 summarizes the work of this paper and explains the next research directions.

## 2. Related Work

In this section, we present the current research related to single gesture recognition, continuous gesture recognition, and sign language gesture recognition in terms of Inertial Measurement Unit (IMU) sensors, vision, and acoustic. A single gesture is the execution of one action at a time, and a continuous gesture is the execution of multiple actions at a time. Additionally, a sign language gesture is the execution of all the gestures contained in a complete sentence at a time.

IMU sensor: IMU sensor is composed of a gyroscope (GYRO) and an accelerometer (ACC). It is usually placed on the user’s arm to capture the movement of the arm. The IMU sensor-based recognition of single gestures works as follows. Trong et al. [15] used the accelerometer and gyroscope in a smartwatch to collect data and combined a one-dimensional convolutional neural network with a bi-directional long short-term memory (1D-CNN-BiLSTM) to analyze and learn the signal features from the sensor signals. The proposed model could achieve a 90% correct rate. Rinalduzzi et al. [16] proposed a machine learning method in conjunction with a magnetic positioning system for recognizing the static gestures associated with the sign language alphabet. The proposed machine learning method is based on a support vector machine, which demonstrated good generalization properties and resulted in a classification accuracy of approximately 97%. There is no related work on continuous gesture recognition, but more on recognition of sign language gestures based on IMU sensors. Hou et al. [17] designed the SignSpeaker system using the IMU sensor of a smartwatch. The SignSpeaker system provided an isolated fine-grained fingerspelling recognition model and a continuous sign language recognition model. Additionally, the system used LSTM and CTC to recognize sign language gestures, but it could not use a smartwatch to recognize two-handed movements. In a sensor-based system, gesture behavior is captured by the wearable sensor. Although it can accurately capture fine-grained behavior characteristics, wearable sensors will bring great inconvenience to daily life, and the cost is high, which can only be used in a few fixed places.

Vision: vision-based systems typically use optical cameras to capture human behavioral features. After the research, vision-based technologies are mainly used to implement continuous gesture and sign language recognition. For continuous gestures, Liu et al. [18] proposed a few-shot continuous gesture recognition scheme based on RGB video. The scheme used Mediapipe to detect the key points of each frame in the video stream, decomposed the basic components of gesture features based on certain human palm structures, and then extracted and combined the above basic gesture features by a lightweight autoencoder network. Mahmoud et al. [19] presented a robust deep learning approach for characterizing, segmenting, and classifying isolated and continuous gesture sequences using depth, RGB, and grayscale input data. The proposed process was suitable for both full human action and gesture recognition. For sign language recognition and sign language translation work, Guo et al. [20] proposed a hierarchical-LSTM framework for sign language translation, which builds a high-level visual semantic embedding model for SLT. However, unseen sentence translation was still a challenging problem with limited sentence data and unsolved out-of-order word alignment. Tang et al. [21] proposed a graph-based multimodal sequential embedding graph (MSeqGraph) network to solve sign language translation with multimodal cues. Experiments on two benchmarks demonstrated the effectiveness of the proposed MSeqGraph and showed that exploiting multimodal cues contributes to a better representation and improved performance. GEN-OBT [22] was proposed to solve the task of sign language translation. Additionally, it designed a CTC-based reverse decoder to convert the generated poses backward into glosses, which guaranteed semantic consistency during the processes of gloss-to-pose and pose-to-gloss. Vision-based sign-language-recognition technology is already mature, and the technology not only considers sign language movements but also incorporates facial expressions, lip-synthesis, etc., which has improved recognition accuracy to a certain extent. Additionally, many sign language translation efforts have been proposed in order to reduce the differences between natural language and sign language recognition. However, the technology is susceptible to light, some infringement of user privacy, and high energy demand for long-term monitoring.

Acoustic: acoustic-based systems typically use speakers and microphones embedded in electronic devices such as smartphones, headphones, and smart bracelets to obtain gesture information. Acoustic gesture recognition can solve the problem of wearable sensors inconvenient high cost but also based on the visual sensitivity to light, the user privacy impact of the problem. Acoustic technology only requires the use of speakers and microphones embedded in smart devices to collect data, reducing device collection costs, expanding the scope of everyday use, and slowing propagation characteristics to enable more accurate recognition. Some recent research works on acoustic gesture recognition have appeared. For single gestures, Mao et al. [23] proposed a system to measure the propagation distance and angle of arrival (AOA) of reflected signals using a four-element microphone array and dual speakers. The system did not allow for finger-level gesture recognition because the user need to hold the phone. Wang et al. [24] solved the frequency selective fading problem caused by multipath effects by periodically transmitting acoustic signals of different frequencies. Additionally, they solved the challenge of insufficient data by automatically generating data based on the correlation between CIR measurements and gesture changes, achieving a breakthrough in the limitations of acoustic gesture recognition in terms of accuracy and robustness. However, this research work can only recognize single gestures and can not handle the case of continuous gestures. For continuous gestures, FingerIO [25] analyzed the echo signal changes caused by finger movements by transmitting orthogonal frequency division multiplexing (OFDM) modulated acoustic signals to achieve accurate tracking of moving objects. However, it only captured finger movements in the two-dimensional plane and could not capture arm movements. The work most similar to ours is the work of Jin’s team. Jin et al. [26] used the speaker and microphone in a commercial headset to send and receive signals for real-time dynamic recognition of sign language gestures, and the system achieved 93.8% recognition for 42 words and 90.6% recognition for 30 sentences. However, the system is dependent on a wearable device (headset) to operate, making it a poor experience to use. Unlike Jin’s team, we did not rely on any wearable device and proposed the first acoustic-based Chinese continuous gesture and sign language recognition system with state-of-the-art results.

## 3. System Design

### 3.1. Overview

The system proposed in this paper is divided into four main parts: data collection, data pre-processing, feature extraction and gesture classification, and the system flow is shown in Figure 1. In the data collection and processing phase, two speakers are used as transmitters to send a single 20 kHz audio signal, a microphone is used as a receiver, and the receiving device records and stores the original echo signal. The raw echo signal is processed and converted to Doppler shift. Firstly, the images are filtered using a Butterworth bandpass filter and STFT, followed by a Gaussian filter to smooth the images. Finally, the dataset is expanded using GAN. In the feature extraction phase, the features of the spectrogram are extracted using the Resnet34 algorithm to generate feature vectors. The gesture classification phase feeds feature vectors into a Bi-LSTM network for classification and recognition. For the sequence prediction problem where the input and output sequences of continuous gestures and sign language gestures are of inconsistent length and difficult to align, we add the CTC algorithm after the Bi-LSTM network, which can convert the feature vector into an indeterminate length gesture sequence or sign language sequence.

### 3.2. Data Collection and Pre-Processing

Data collection and pre-processing. The frequency of living noise is usually located at [1000, 4000] Hz [27]. In order to ensure that the signal frequency used in the experiment does not conflict with the frequency of living noise, this paper sets the speaker to send a single audio signal of 20 kHz. The single audio signal has the advantage of low complexity and high resolution in terms of Doppler shift [28]. Figure 2, Figure 3 and Figure 4 show the schematic diagrams of the Doppler effect corresponding to 15 single gestures, six sets of continuous gestures, and six sets of sign language gesture data after pre-processing, respectively. To better describe the gesture under test, in Figure 2 we use X→ to indicate the hand motion along the X-axis and double arrows (e.g., X ↔) to indicate the back and forth motion of the hand along the X-axis.

Hand gesture data processing. A Butterworth bandpass filter with a frequency of [19,000, 21,000] Hz is first used to eliminate the interference of background noise, followed by an STFT to extract the Doppler shift caused by the gesture motion. STFT is the most commonly used method for time-frequency analysis, but the time resolution and frequency resolution are difficult to balance. To balance real-time and frequency resolution, we set the frame length to 8192 and the window step size to 1024. The frequency change of the signal after reflection is estimated by calculating the Doppler shift, and the image shown in Figure 5a is obtained.
(1)$$\Delta f=f_{0}\times \left|1-\frac{v_{s}\pm v_{f}}{v_{s}\mp v_{f}}\right|$$
where $f_{0}$ is the frequency of the signal sent by the speaker (20 kHz), $v_{s}$ is the speed of sound (340 m/s), $v_{f}$ is the speed of gesture movement (maximum movement speed 4 m/s). So the synthesized frequency shift is about 470.6 Hz, and the effective frequency range should be within [19,530, 20,470] Hz.

To eliminate the effect of isolated noise generated by sudden hardware noise on the signal, the point where the STFT value changes most dramatically, 0.15, is set as the threshold value, and any isolated noise less than this threshold is set to 0. After we use a Gaussian filter to smooth the image. For two-dimensional images, the following Gaussian functions are used for smoothing.
(2)$$G\left(x,y\right)=\frac{1}{2\pi \sigma^{2}}exp\left(-\frac{x^{2}+y^{2}}{2\sigma^{2}}\right)$$
where *x* is the distance of the horizontal axis from the origin, *y* is the distance of the vertical axis from the origin, σ is the standard deviation of the Gaussian distribution, and the processed image is shown in Figure 5b.

### 3.3. Data Augmentation

Traditional data augmentation [29] generates new data from limited data by synthesis or transformation. Traditional data augmentation techniques in the image domain are based on a series of known affine transformations, such as rotation, scaling, displacement, etc., and some simple image processing tools, such as light color transformation, contrast transformation, noise addition, etc. This method of data augmentation based on geometric transformation and image manipulation can alleviate the overfitting problem of neural networks and improve the generalization ability to a certain extent, but the addition of new data does not fundamentally solve the problem of insufficient data compared with the original data. The recent emergence of GAN [30] can also be used for data augmentation. This network-based synthesis method is more complex than traditional data enhancement techniques, but the generated samples are more diverse and can be applied to various scenarios, such as image editing and image denoising.

GAN consists of a discriminator network and a generator network. Discriminators are two-category classification networks that distinguish whether *x* comes from the true distribution or the generative model. Unlike the fully connected neural network-based discriminator in the original GAN network, we use CNN as a discriminator to better extract features in gesture images. The generation needs to make the discriminator network distinguish its own generated samples from real samples. First, the generator randomly initializes a latent vector, and then continuously performs convolution and upsampling operations to transform the latent vector to the size of the actual image. The basic structure of GAN is shown in Figure 6.

*X* represents the real data, *z* represents the noise of the generator network, G(z) means unreal data generated by the generator network, and D(x) represents the probability that *x* belongs to the real sample distribution, where D∈[0,1]. The optimization principle of GAN is simply that the generator network, *G*, generates G(z) through continuous training and learning and makes the discriminator network, *D*, unable to distinguish the difference between G(z) and *X*. *D* is to improve their discriminant ability through continuous training and learning, that is, to recognize that *X* and G(z) are different.

The optimization function of the whole GAN network can be summarized by Equation (3):(3)$$\min_{G}\max_{D}V\left(D,G\right)=E_{x∼P_{data}\left(x\right)}\left[logD\left(x\right)\right]+E_{Z∼P_{Z}\left(Z\right)}\left[log\left(1-D\left(G\left(z\right)\right)\right)\right]$$

The main meaning of this equation is that one is the *G* remains constant and the *D* wants to distinguish the real samples from the training samples. Additionally, the other is the *D* remains constant and by adjusting the *G* it wants the *D* to make a mistake and not let it distinguish as much as possible. The training process of generators and discriminators is iterated alternately. First, optimize the discriminator *D*. The purpose of the discriminator is to be able to correctly distinguish between G(z) and *X*. When optimizing the discriminator network, it is necessary to give *D* and *G* in advance and try to increase D(x) and decrease D(G(z)), i.e., the optimization objective of the discriminator network is $\max_{D}V\left(D,G\right)$. When optimizing the generator network, it is also necessary to give *D* and *G* in advance and optimize $\min_{G}V\left(D,G\right)$.

Specifically, we set the set of input images $P=\{p_{1},p_{2},\ldots ,p_{m}\}$. To train the discriminator model, for each small batch, *m* samples are sampled from the prior noise distribution $p_{g}\left(z\right)$ as $\left\{z^{\left(1\right)},\ldots ,z^{\left(m\right)}\right\}$, and *m* samples are obtained from the real data distribution $p_{data}\left(x\right)$ as $\left\{x^{\left(1\right)},\ldots ,x^{\left(m\right)}\right\}$, and the discriminator is updated by boosting the random gradient Equation (4). When training the generator model, for each small batch, again *m* samples are sampled from the prior noise distribution $p_{g}\left(z\right)$ and the generator is updated by reducing the random gradient Equation (5).
(4)$$\nabla_{\theta_{d}}\frac{1}{m}\sum_{i=1}^{m}\left[logD\left(x^{\left(i\right)}\right)+log\left(1-D\left(G\left(z^{\left(i\right)}\right)\right)\right)\right]$$
(5)$$\nabla_{\theta_{g}}\frac{1}{m}\sum_{i=1}^{m}log\left(1-D\left(G\left(z^{\left(i\right)}\right)\right)\right)$$

In practice, we build a GAN network for each category of data separately. As shown in Figure 6, the generated images are basically the same as the original images, and it is difficult to distinguish the difference between the real samples and the generated samples. Therefore, by means of GAN, a large amount of high-quality data can be expanded in a short time and used for the training of subsequent gesture recognition models.

### 3.4. Feature Extraction and Gesture Classification

#### 3.4.1. Feature Extraction

In this paper, we use ResNet34 [31] to extract features, and its structure is shown in Figure 7. The ResNet34 model has 34 convolutional layers, including a total of 16 residual learning units, where all convolutional operations use a convolutional kernel of size 3 × 3. The spectrogram obtained from data augmentation is used as the input to ResNet34, ensuring that the input images are all 64 × 64 pixels in size. After each convolutional layer and before the activation function (ReLU), batch normalization is used to accelerate the convergence. Performing reshapes and flatten operations on the output of the last residual block, we can obtain the feature vector $y=[y_{1},y_{2},\ldots ,y_{T}]$, the total number of feature vectors T=512, and the length of each feature vector is 16.

#### 3.4.2. Gesture Classification

Bi-LSTM. Traditional LSTM can only encode information from front to back, not from back to front, but information from back to front is also important for determining activity. Bi-LSTM [32] can better capture the semantic dependencies in both directions. The Bi-LSTM network computation is usually divided into the following four steps:

Step 1: from the forgetting gate $f_{t}$, determine the information to be discarded from the cell state. The forgetting gate can read the output $h_{t-1}$ of the previous sequence, the input $x_{t}$ of the current sequence and perform the Sigmoid operation:(6)$$f_{t}=\sigma \left(W_{f}\cdot \left[h_{t-1},x_{t}\right]+b_{f}\right)$$

Step 2: determine what new information will be stored in the sequence state. First of all, the Sigmoid layer determines which values we will update. Subsequently, a new vector of candidate values ${\tilde{C}}_{t}$ is created using the tanh layer.
(7)$$i_{t}=\sigma \left(W_{i}\cdot \left[h_{t-1},x_{t}\right]+b_{i}\right)$$
(8)$${\tilde{C}}_{t}=tanh\left(W_{c}\cdot \left[h_{t-1},x_{t}\right]+b_{c}\right)$$

Step 3: update sequence status.
(9)$$C_{t}=f_{t}*C_{t-1}+i_{t}*{\tilde{C}}_{t}$$

Step 4: determine the output values based on the updated sequence states. First of all, the Sigmoid layer is used to determine which sequence states can be output. Then the sequence states $C_{t}$ obtained in the third step are mapped to between −1 and 1 using tanh and multiplied with the Sigmoid gate $o_{t}$ to obtain the final output $h_{t}$.
(10)$$o_{t}=\sigma \left(W_{0}\cdot \left[h_{t-1},x_{t}\right]+b_{0}\right)$$
(11)$$h_{t}=o_{t}*tanh\left(C_{t}\right)$$
where $h_{t-1}$ denotes the output of the spectrogram sequence at the previous moment, $x_{t}$ denotes the input of the spectrogram sequence at the current moment, *W* and *b* are the weight term and bias term to be learned, respectively, σ denotes the Sigmoid operation, $f_{t}$ denotes the output of the forgotten gate at time *t*, $i_{t}$ denotes the information of the spectrogram sequence to be activated at the moment *t*, $C_{t-1}$, and $C_{t}$ denote the state of the spectrogram feature sequence at the moment t−1 and moment *t*, respectively, $h_{t}$ is the output result of the output gate at time *t*.

Specifically, the feature vectors *y* extracted by the residual neural network are passed to two LSTM layers, each of which has T(T=512) LSTM storage units. To improve the generalization ability of the model set the dropout of the model to 0.8. These two layers perform sequence feature extraction in opposite directions, and each LSTM memory cell will be computed by three gating units. After calculation, the output $H_{forward}$ of the forward LSTM and the output $H_{backward}$ of the reverse LSTM can be obtained. After that, we concatenate and flatten $H_{forward}$ and $H_{backward}$ to obtain the vector *P*. In the single-category gesture recognition task, since the classifier eventually needs to recognize 15 gestures, we design a fully connected neural network with 15 output neurons. Finally, softmax operations are performed on the output of the fully connected layer to accurately classify and recognize different gestures. In the case of continuous gesture or sign language recognition tasks, it is necessary to input the vector *p* to the CTC algorithm for processing, and we will describe this process in detail in the next section.

CTC. In this paper, we use the CTC [33] algorithm as a classifier for the continuous gesture and sign-language-gesture recognition. CTC is an algorithm commonly used in speech recognition, text recognition, and other fields to solve the problem of unaligned input and output sequences of different lengths. Unlike single gesture prediction, after the Bi-LSTM network obtains the feature vector $p\in R^{c\times n}$ (*c* represents the length of the feature vector and *n* represents the number of classes of gestures or sign language), the fully connected layer is no longer designed, but *p* is input into the CTC algorithm. Algorithm 1 shows the steps of the CTC method.

First, the CTC layer receives the output sequence *p* from the Bi-LSTM and then computes the probability $p_{ctc}\left(Y|p\right)$ between *p* and the true label *Y* on any alignment π, where π[t] is the character ID aligned to the *t*th frame in *p*, as follows:(12)$$C=softmax(pW^{ctc}+b^{ctc})$$
(13)$$p\left(B\left(\pi \right)=Y|p\right)=\prod_{t=1}^{n^{sub}}C\left[t,\pi \left[t\right]\right]$$
(14)$$p_{ctc}\left(Y|p\right)=\sum_{\pi^{{}^{\prime }}\in B^{-1}\left(Y\right)}p\left(B\left(\pi \right)=Y|p\right)$$
where $W^{ctc}\in R^{n\times char}$ and $b^{ctc}\in R^{char}$ are learnable parameters, $C\in R^{c\times char}$ is the output of CTC, C[t,π[t]] is the probability that the output character π[t] is aligned with the *t*th frame. The many-to-one mapping B(π) is used to remove redundant symbols from the alignment π, for example, B(aaØb)=ab, where Ø is a blank character and the one-to-many mapping $B^{-1}$ projects the sequence of characters into a set of character sequences with redundant symbols.
(15)$$B^{-1}\left(Y\right)=\left\{\pi |Y=B\left(\pi \right)\right\}$$

In the training phase, we train the entire set of models using the CTC loss function.
(16)$$L_{ctc}=-log_{ctc}\left(Y|p\right)$$

In the prediction phase, we need to use the Beam Search Decoding algorithm to convert the feature vectors predicted by Bi-LSTM into the final sign language sequence prediction results. In the sequence prediction problem, the model prediction process is essentially a spatial search process, the core of which is to calculate the probability of expanding nodes at each step. The sequence with the highest probability the last time is taken as the final output of the model.
**Algorithm 1** Steps of CTC**Input:** Sequence of strings *L*, Number of nodes in each expansion *W***Output:** The sequence *Q* with the maximum probability at time *T*1: **for** t=1 to *T* **do**2:       Set $\hat{B}$ = the *W* most probable sequences in *B* (*L* when t=1)3:       Set B={ }4:       **for** $p\in \hat{B}$ **do**5:             **if** p≠Ø **then**6:                   $r^{+}\left(p,t\right)=r^{+}\left(p,\;t-1\right)y_{p^{e}}^{t}$7:                   **if** $\hat{p}\in \hat{B}$ **then**8:                         $r^{+}\left(p,\;t\right)+=Probability\left(p^{e},\;\hat{p},\;t\right)$9:             $r^{-}\left(p,\;t\right)=r\left(p,\;t-1\right)y_{b}^{t}$10:             add *p* to *B*11:             **for** k=1 to *K* **do**12:                   $r^{-}\left(p+k,\;t\right)=0$13:                   $r^{+}\left(p+k,\;t\right)=Probability\left(k,\;p,\;t\right)$14:                   add (p+k) to *B*15: **return** 
$\underset{p\in B}{argmax}\;r{\left(p,\;T\right)}^{\frac{1}{\left|p\right|}}$

## 4. Experimentation and Evaluation

### 4.1. Experiment Setting

Experimental platform. In the experimental phase, ASDP equipped with one microphone and two speakers were chosen as the data collection tool. Two speakers are transmitters (Tx) and one microphone is a receiver (Rx). ASDP is an acoustic software-defined radio platform, a multi-functional communication and sensing platform. The ASDP is mainly composed of hardware, such as Raspberry Pi, INMP411, TPS54332, WM8731, etc. The platform is shown in Figure 8a. Set the speaker to emit a 20 kHz continuous single audio signal and set the microphone sampling rate to 44.1 kHz.

Dataset. We collected data in two scenarios, laboratory, and corridor, and the real scenario was shown in Figure 8b,c. We invited 6 male volunteers and 6 female volunteers to perform 15 single gestures. Additionally, we collected 720 sets of data under 4 practical influencing factors of distance, speed, noise, and angle. Then we invited 2 male volunteers and 2 female volunteers to perform 6 continuous gestures and 6 sign language gestures, and 120 sets of data were collected for each. All of the above actions were performed by the volunteer while keeping the body stationary and within a distance of 0.2 m to 0.5 m from the device. The open source address for the dataset is: https://github.com/yuejiaowang/database (accessed on 31 December 2022).

Implementation details. In our experiments, the input image for single gesture recognition is resized to 512 × 512, and the input image for the continuous gesture and sign language gesture recognition is resized to 620 × 462. For data augmentation, we use the method mentioned in Section 3.3 for 20× data augmentation with the addition of random scaling and random rotation. In the experiments for single gesture recognition, continuous gesture recognition, and sign language recognition, we use 80% of the data as the training set and the remaining 20% as the test set. Additionally, the results reported in the experiments are all 5-fold cross-validation results. Our network architecture is implemented in PyTorch. In single gesture recognition experiment, we use Adam optimizer with a learning rate $1\times 10^{-3}$ and set the batch size to 16. A total of 60 epochs are trained. In the experiments of continuous gesture and sign language gesture, we use the Adam optimizer with an initial learning rate of $1\times 10^{-4}$ and set the batch size to 2. A total of 100 epochs are trained and the learning rate is reduced by a factor of 10 in the 60th and 80th epochs, respectively. All recognition models are not loaded with any pre-training weights and experiments are conducted on NVIDIA Tesla P40 GPU.

### 4.2. Ablation Study

#### 4.2.1. Impact of Different Influencing Factors

In order to evaluate the UltrasonicGS method in terms of different influencing factors, this paper designed experiments in three aspects: distance between gesture and transceiver, angle of arrival, and gesture speed in laboratory and corridor environments, respectively. (1) Five experimenters were asked to execute the gesture at 5 cm, 15 cm, 25 cm, 35 cm, and 50 cm from the transceiver position. (2) Five experimenters were asked to execute the gestures at 30°, 60°, 90°, 120°, and 150° with the equipment. (3) Five experimenters were asked to perform gestures of duration 0.5 s, 1 s, 1.5 s, 2 s, and 2.5 s, respectively. The results of the experiment are shown in Figure 9.

Figure 9a shows the impact of environment and distance on the correct gesture recognition rate. From the perspective of the environment, it can be seen that the recognition result of the corridor environment is higher than that of the laboratory environment at the same distance from the transceiver. This is due to the fact that the laboratory contains regularly distributed equipment with tables and chairs, so the multipath effect is more disturbing. From the perspective of distance, it can be seen that when the distance between the hand and the device is 15 cm, the correct gesture recognition rate reaches up to 98%. As the distance between the hand and the device increases, the correct gesture recognition rate gradually decreases. When the distance is 50 cm, the correct gesture recognition rate is close to 88%. The reason for this phenomenon is that when the distance is too small, the signal reflected by the hand is not completely received by the microphone. When the distance is too large, the interference of the multipath effect on the reflected signal increases.

Figure 9b shows the impact of environment and angle of arrival on the correct gesture recognition rate. As can be seen from the figure, when the experimenter performs the gesture at 90° to the device, the gesture recognition rate is 99% correct. When the experimenter is at 30°, 60°, 120°, and 150° to the device, the gesture recognition rate does not differ much, fluctuating around 96%. This is because when the angle of arrival is 90°, the direction of hand motion is perpendicular to the signal domain, which has a greater impact on the signal. Additionally, when the experimenter is at other angles to the device, the hand motion generates a horizontal motion component with a smaller signal amplitude. Overall, UltrasonicGS is able to maintain high performance specifications in all directions.

Figure 9c shows the impact of environment and speed on the correct rate of gesture recognition. The figure shows that when the gesture duration is 1.5 s, the highest correct gesture recognition rate can reach 98.7%. As the duration of the gesture increases or decreases, the correct gesture recognition rate decreases. This is because the gesture duration is too long, the gesture speed is too slow, and the signal change caused by the Doppler shift is not obvious. The gesture duration is too short, the gesture speed is too fast, and the microphone fails to receive the complete signal in a short period.

The experimental results demonstrate that UltrasonicGS maintains good recognition performance within a distance of 50 cm between the hand and the transceiver, in all directions, and within a hand gesture duration of 2.5 s.

#### 4.2.2. Impact of Noise and Personnel Interference

To evaluate the impact of the UltrasonicGS method on ambient noise, line-of-sight (LOS), non-line-of-sight (NLOS), and personnel interference, we designed the following two experiments. (1) Experimenters were asked to perform 15 gestures at 15 cm from the device position in the no noise, low-frequency noise, and 19 kHz ultrasonic noise of LOS and NLOS environments, respectively. (2) Experimenters were asked to perform 15 gestures in four situations of interference: no human interference, human static interference (experimenter standing still), human light interference (experimenter walking back and forth), and human heavy interference (experimenter executing disturbance gestures while walking).

The results in Figure 10a show that the correct gesture recognition rate stays above 98% in the LOS environment and fluctuates around 91.2% in the NLOS environment. This is due to the better signal quality and higher throughput in the LOS channel model, however, the multipath effect in the NLOS channel model leads to frequency selective fading. From the perspective of noise, it can be seen that low-frequency noise and ultrasonic noise have basically no effect on the experimental results, which further verifies that the data pre-processing method proposed in this paper can remove noise interference well.

The cumulative distribution functions (CDF) of the error rate for different interference states are given in Figure 10b. The x-axis represents the recognition error rate and the y-axis represents the CDF percentage. At a CDF of 0.8, the error rates corresponding to no human interference, human static interference, human light interference, and human heavy interference are 0.09, 0.11, 0.14, and 0.18, respectively. The highest accuracy is achieved in an environment without human interference, and the worst recognition performance is achieved in an environment with human heavy interference. However, the error rate of about 80% of the test data is less than 18%, which indicates that the method proposed in this paper has some anti-interference capability.

#### 4.2.3. Impact of Dataset Size

To evaluate whether data augmentation helps to improve the performance of the gesture recognition model, we conducted experiments in three tasks: single gestures, continuous gestures, and sign language gestures, respectively. Figure 11 shows the ROC curves with and without data augmentation in turn.

In Figure 11, the blue curve and the area surrounded by the x-axis are the Area Under Curve (AUC) when the data augmentation method is used in the UltrasonicGS method and the red curve and the area surrounded by the x-axis are the AUC when the data augmentation method is not used. We can observe that, whether it is a single gesture, continuous gesture, or sign language gesture, when we use the GAN data augmentation method, the receiver operating characteristic (ROC) curve rises faster and the area occupied by AUC will be larger, and the recognition effect will be better than without the method. Therefore, data augmentation techniques can extend the dataset and help to improve the performance of the gesture recognition model. We will use data augmentation techniques in a series of subsequent experiments.

### 4.3. Comparison with the State-Of-The-Art Methods

In order to verify the superiority of our proposed method in gesture recognition, we compared it with the classical methods of acoustic sensing gesture recognition in recent years. Table 1 details the differences between the five methods with respect to the five aspects of sending signal, device, application, algorithm, and feature extraction for the word level. Table 2 compares with SonicASL, which is based only on acoustics for sign language sensing.

In Table 1, it can be observed that the recognition accuracy of our proposed method reaches 98.8%, which is the best performance among all methods. AudioGest and SoundWave are suitable for recognizing whole-hand gestures, while our dataset contains fine-grained finger-level gestures, resulting in poor recognition of the above two methods, with recognition accuracies of 89.1% and 88.6%, respectively. Thanks to the multiscale semantic features extracted by our CNN fed into the Bi-LSTM algorithm, we can make the classification network fuse the information of feature dimension and temporary dimension. Additionally, the recognition performance is significantly better than that of other finger-level recognition methods UltraGesture and Push. In Table 2, both SonicASL and our method can recognize word-level and sentence-level gesture activities. Additionally, our proposed method recognizes individual gestures with a 5% higher correct rate than SonicASL but recognizes sign language gestures with 4.3% lower than the comparison method. The reason for this situation is that we perform Chinese sign language recognition, while SonicASL performs English sign language recognition, which is a more complex situation involving homophones and split words. After experiments, our method increases the recognition correct rate when recognizing continuous sentences in English. Therefore, our proposed method can meet the demand for action recognition in general perceptual space and can ensure stable recognition accuracy.

### 4.4. Overall Performance

#### 4.4.1. Overall Accuracy of Single Gestures

In order to evaluate the accuracy of 15 single gestures, the experimenters were asked to perform this experiment in different environments (multipath-rich and multipath-not-rich rooms) and with different influencing factors (distance angle and speed) in this section. The results of the experiment are shown in Figure 12.

Figure 12 shows the overall confusion matrix for performing 15 single gestures in different environments and with different influencing factors. The results of the confusion matrix show that the UltrasonicGS method has a combined recognition rate of 98.8%. Among them, 10 gestures, such as “1, 2, pinch, pull, push” can achieve 100% correct recognition rate. In order to ensure the authenticity and expandability of the dataset, each experimenter can perform the gestures “3” and “OK” according to their own habits when actually collecting data. This resulted in similar gestures for “3” and “OK”, with a small difference in the Doppler effect. The recognition rate of the above two gestures is slightly lower, but the correct rate is 93%. In summary, the UltrasonicGS method is able to distinguish the 15 single gesture actions well.

#### 4.4.2. Performance Evaluation of Continuous Gesture

To evaluate the performance of the UltrasonicGS method for continuous gesture recognition, four classification models were selected. ResNet34 extracted feature values, Bi-LSTM, and CTC-classified gestures. VGG16 [37] extracted feature values, Bi-LSTM and CTC classified gestures. ResNet34 extracted feature values, LSTM [38], and CTC classified gestures. VGG16 extracted feature values, LSTM, and CTC-classified gestures. The six groups of continuous gestures selected in the experiment were: Spread and Pinch; Push and Pull; Hover and OK; Around Left and Around Right; One, Two, and Three; and Slide Up, Slide Down, Slide Left, and Slide Right. The experimental results are shown in Figure 13 and Figure 14.

The CDF of error rates for different classification algorithms are given in Figure 13. The six CDF figures represent six different continuous gestures, where the first four CDF figures are continuous gestures composed of two gestures, the fifth is a continuous gesture composed of three gestures, and the sixth is a continuous gesture composed of four gestures. Globally, the six CDF plots of error rates for each classification algorithm vary essentially uniformly. Using ResNet34 to extract feature values, Bi-LSTM and CTC achieve the highest accuracy for classification of continuous gestures, where approximately 89% of the tested data have an error rate of less than 10%. Using ResNet34 to extract feature values, LSTM and CTC gesture classification have similar recognition rates as using VGG16 to extract feature values, with approximately 80% of the test data having an error rate of less than 20%.

Figure 14 shows the accuracy of six continuous gestures with different classification models. C1, C2, C3, C4, C5, and C6 correspond to each of the six gestures in Figure 13. For each gesture using ResNet34 to extract the feature values, both Bi-LSTM and CTC classification achieved the highest accuracy, with an average accuracy of 92.4%. Using VGG16 to extract the feature values, LSTM and CTC achieved the lowest accuracy, with an average accuracy of 90.97%. This shows that the method used in this paper can recognize not only single gestures but also continuous gestures. Additionally, the method incorporates the information of feature dimension and temporary dimension, which effectively improves the accuracy of gesture recognition.

#### 4.4.3. Performance Evaluation of Sign Language Gesture

In order to evaluate the performance of the UltrasonicGS method for sign language gesture recognition, we also chose the same four classification models as in the previous experimental continuous gesture performance evaluation for “I am a teacher.” “I am fine, thanks.” “What day is today?” “Sorry, I am late.” “What do you do?” “What is your name?” six groups of Chinese sign language carried out the experiment, and the experimental results are shown in Figure 15 and Figure 16.

The ROC curves of different classification models are given in Figure 15. The x-axis represents the false positive case rate, the y-axis represents the true case rate, and the six ROC plots represent the six different sign language gestures. Globally, there is almost no difference in ROC curves and similar AUC areas for the six different sentence descriptions, which indicates that the same model is similarly effective in recognizing six different sets of sign language sentences. Using ResNet34 to extract feature values, the Bi-LSTM and CTC algorithms are used to classify sign language gestures with the fastest ROC curve change and the largest AUC area, while the other three classification models have a slightly slower ROC curve change and smaller corresponding AUC areas.

Figure 16 shows the accuracy of the six sign language gestures under different classification models. S1, S2, S3, S4, S5, and S6 correspond to the six gestures in Figure 15. For each gesture using ResNet34 to extract the feature values, both Bi-LSTM and CTC classification achieved the highest accuracy with an average accuracy of 86.3%. Using VGG16 to extract feature values, LSTM and CTC achieved the lowest correct classification rate of 84.2% for gestures. This shows that the method used in this paper can recognize not only continuous gestures but also sign language gestures. The method incorporates the information on feature dimension and temporary dimension, which effectively improves the accuracy of gesture recognition.

## 5. Conclusions

In this study, we propose the UltrasonicGS, a highly robust gesture and sign language recognition method based on ultrasonic signals. The method can recognize 15 single gestures with high accuracy and robustness. Additionally, in order to satisfy more audience groups, especially special groups, such as the deaf, we extend the method to recognize continuous gestures and sign language gestures. To achieve fine-grained gesture recognition, the extraction of feature values using ResNet34 and the classification of single gestures by Bi-LSTM. For continuous gestures and sign language gestures, we add CTC algorithm after Bi-LSTM network to solve the problem of inconsistent length and difficult alignment of input and output sequences of continuous gestures and sign language gestures. To further improve the robustness of UltrasonicGS, automatic data generation using GAN can alleviate the problem of neural network overfitting and improve the generalization ability to a certain extent. Finally, a dataset containing three categories of gestural behavior is constructed and open sourced. The experimental results show that the method recognize a distance of 0.5m, and the overall correct rate of single gestures reach 98.8%, and the average correct rates of recognition for six groups of continuous gestures and sign language gestures are 92.4% and 86.3%, respectively.

In future work, we will further investigate (1) improving the recognition accuracy of this model for sign language datasets and (2) replacing the collection device with a cell phone to achieve sign language gesture speech conversion and text conversion functions to improve human–computer interaction.

## Figures and Tables

**Figure 1 sensors-23-01790-f001:**
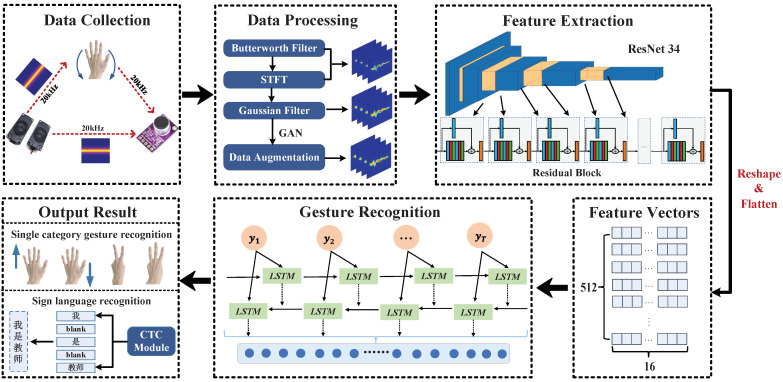
Overview of UltrasonicGS (In the output result module, “我是教师” is a Chinese sentence, which means “I am a teacher” in English. Where “我”“是”“教师” correspond to “I”, “am” and “teacher” respectively).

**Figure 2 sensors-23-01790-f002:**
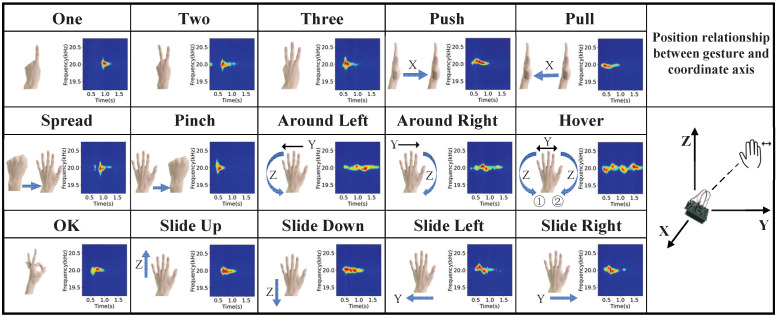
Single gesture spectrogram.

**Figure 3 sensors-23-01790-f003:**
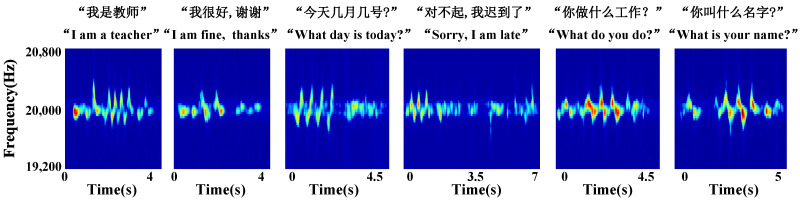
Continuous gesture spectrogram.

**Figure 4 sensors-23-01790-f004:**
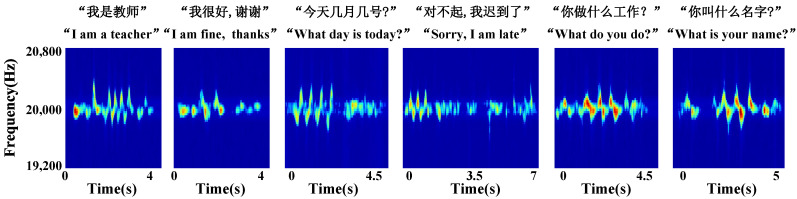
Sign language gesture spectrogram.

**Figure 5 sensors-23-01790-f005:**
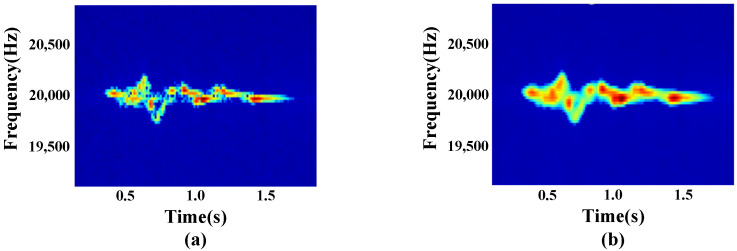
Single gesture action data processing process. (**a**) Bandpass filtering data; (**b**) Gaussian smoothing data.

**Figure 6 sensors-23-01790-f006:**
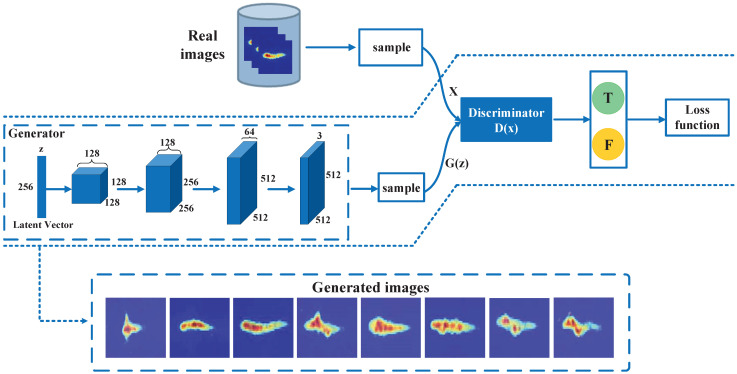
Overview of the GAN.

**Figure 7 sensors-23-01790-f007:**
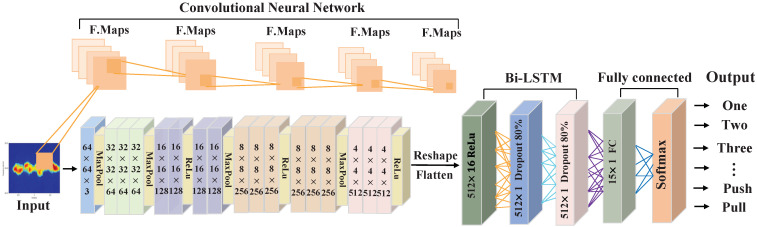
Overview of the ResNet34 and Bi-LSTM.

**Figure 8 sensors-23-01790-f008:**
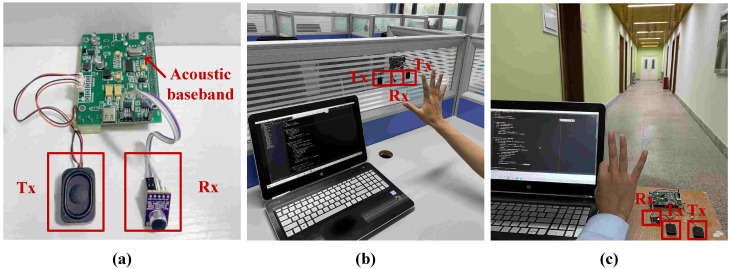
Experimental equipment and environment. (**a**) Data collection equipment; (**b**) Laboratory environment; (**c**) Corridor environment.

**Figure 9 sensors-23-01790-f009:**
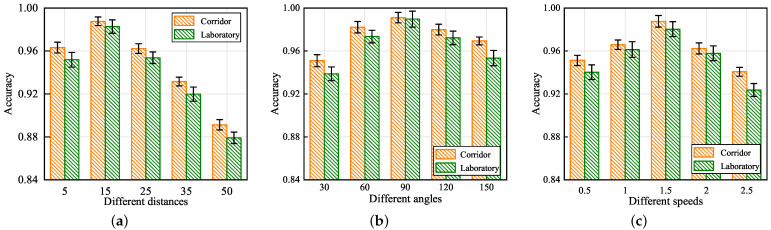
Impact of different distance, angles and speeds. (**a**) Impact of different distance; (**b**) impact of different angles; and (**c**) impact of different speeds.

**Figure 10 sensors-23-01790-f010:**
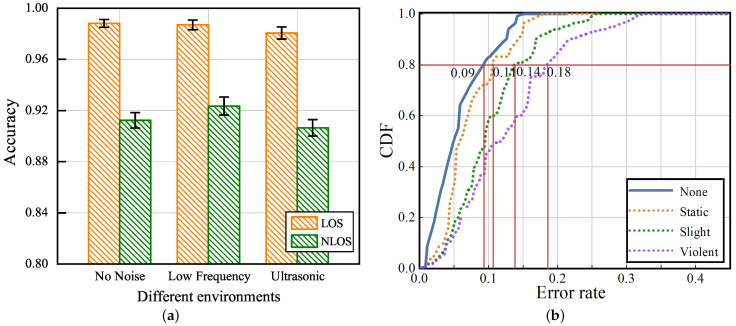
Impact of different environments and interference states. (**a**) Impact of different environments and (**b**) impact of different interference states.

**Figure 11 sensors-23-01790-f011:**
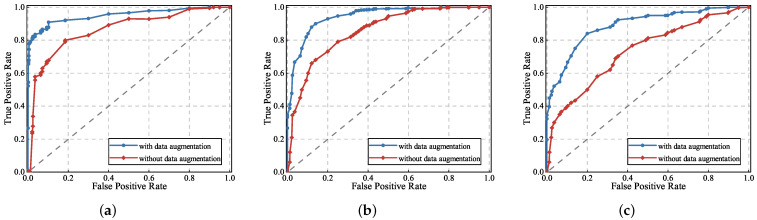
Impact on recognition performance of single gesture, continuous gesture and sign language gesture when data augmentation is used or not. (**a**) Single gesture; (**b**) continuous gesture; and (**c**) sign language gesture.

**Figure 12 sensors-23-01790-f012:**
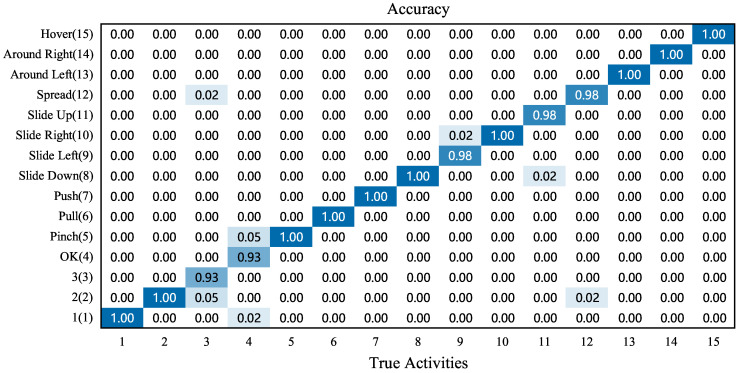
Overall performance of single gestures.

**Figure 13 sensors-23-01790-f013:**
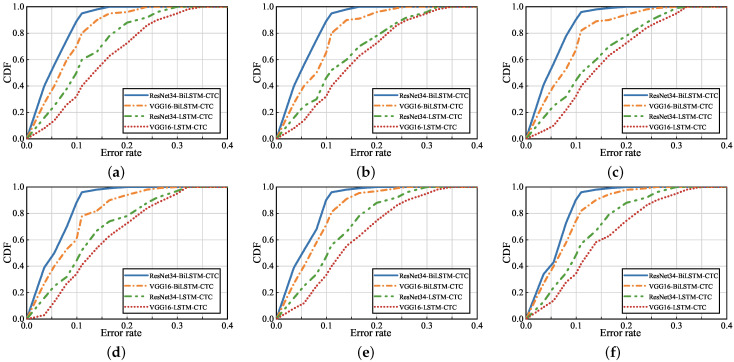
Impact of classification model on continuous gesture performance. (**a**) Spread and Pinch; (**b**) Push and Pull; (**c**) Hover and OK; (**d**) Around Left and Around Right; (**e**) One, Two, and Three; and (**f**) Slide Up, Slide Down, Slide Left, and Slide Right.

**Figure 14 sensors-23-01790-f014:**
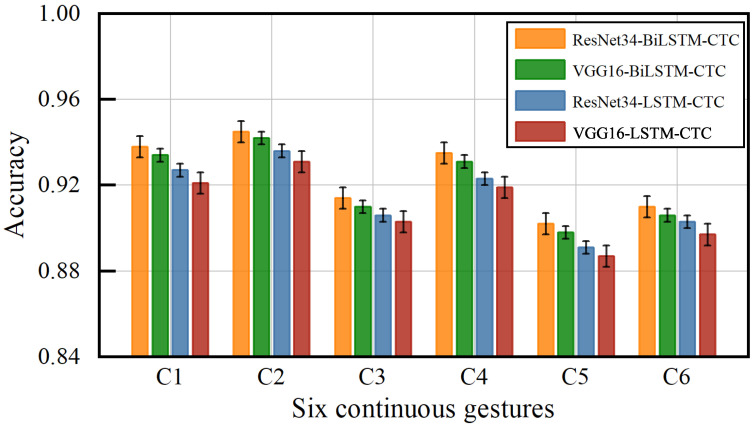
Impact of different models on accuracy of continuous gestures.

**Figure 15 sensors-23-01790-f015:**
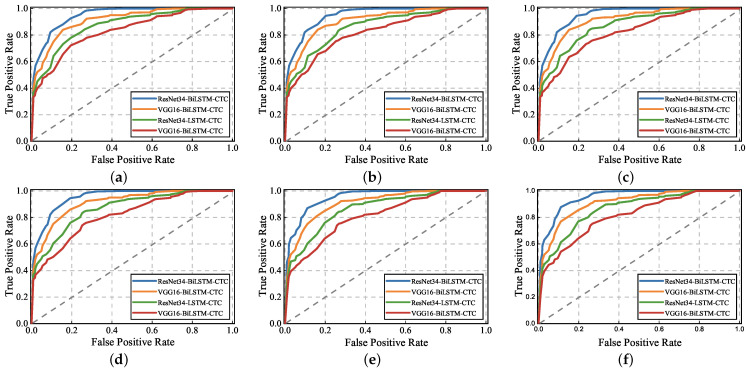
Impact of classification model on sign language gesture performance. (**a**) I am a teacher. (**b**) I am fine, thanks. (**c**) What day is today? (**d**) Sorry, I am late. (**e**) What do you do? (**f**) What is your name?

**Figure 16 sensors-23-01790-f016:**
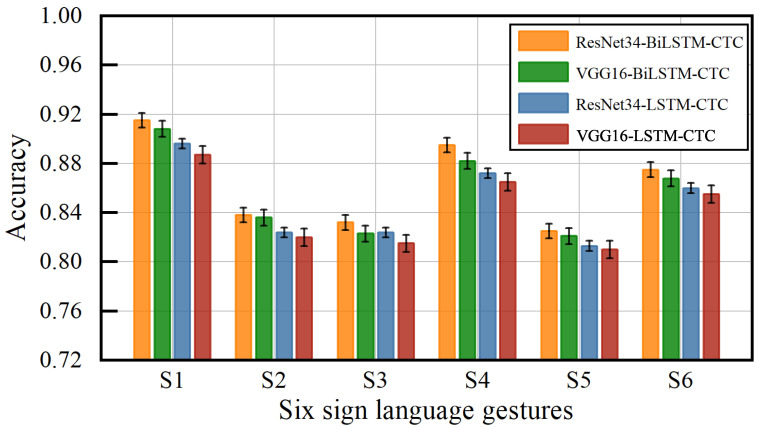
Impact of different models on accuracy of sign language gestures.

**Table 1 sensors-23-01790-t001:** Comparison with the word level methods.

Project	Signal	Device Free	Application	Algorithm	Feature	Accuracy
AudioGest [34]	Ultrasound	Yes	Whole-hand Gesture	/	Doppler Effect	89.1%
SoundWave [35]	Ultrasound	Yes	Whole-hand Gesture	CNN	Doppler Effect	88.6%
UltraGesture [36]	Ultrasound	Yes	Finger-level Gesture	CNN	CIR	93.5%
Push [24]	Ultrasound	Yes	Finger-level Gesture	CNN+LSTM	CIR	95.3%
Ours	Ultrasound	Yes	Finger-level Gesture	CNN+Bi-LSTM	Doppler Effect	98.8%

**Table 2 sensors-23-01790-t002:** Comparison with the sentence level methods.

Project	Signal	Application	Algorithm	Single	Continuous	Sign Language
SonicASL [26]	Ultrasound	Word and Sentence	CNN+LSTM+CTC	93.8%	/	90.6%
Ours	Ultrasound	Word and Sentence	CNN+Bi-LSTM+CTC	98.8%	92.4%	86.3%

## Data Availability

Data is contained within the article.

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
