# Peer review of "UltrasonicGS: A Highly Robust Gesture and Sign Language Recognition Method Based on Ultrasonic Signals"

_sensors, 2023, doi:10.3390/s23041790_

Round 1
Reviewer 1 Report (Previous Reviewer 4)
1. There are many syntax errors in the statement, which should be corrected properly. For example, in the second paragraph of subsection 4.1, "The we invited 2 male volunteers and 2 female....." is incorrect. In the eighth paragraph of subsection 1, "we verify the method has high robustness..." is incorrect.
Author Response
Please see the attachment.

Reviewer 2 Report (New Reviewer)
This manuscript proposes UltrasonicGs, which can recognize single gesture recognition, continuous gestures, and sign language gesture recognition without wearable devices. This manuscript has certain scientific significance, but the content still has some problems. Therefore, in the view of the reviewers, this paper may need to be revised. The specific comments of the reviewer are as follows:
1. May the loss of the amplitude difference in the spectrogram due to the binarization affect the recognition accuracy? May the authors explain the purpose of binarizing the amplitude?
2. As shown in section 3.3: “As shown in Fig. 6, the generated images are basically the same as the original images, and it is difficult to distinguish the difference between the real samples and the generated samples.” May the authors explain how to distinguish the difference between the real samples and the generated samples by some examples? The validity of the generated samples needs to be verified.
3. As Figure 2 shows that the spectrograms of gestures “3” and “OK” are different. May the authors explain why they are similar? Whether the spectrogram cannot reflect the difference clearly?
Author Response
Please see the attachment.

Reviewer 3 Report (New Reviewer)
The theme of the work is interesting, but its contribution is not clear.
There is not much evidence that the proposal is better than the state of the art, Table 2 does not reflect it.
Author Response
Please see the attachment.

This manuscript is a resubmission of an earlier submission. The following is a list of the peer review reports and author responses from that submission.
Round 1
Reviewer 1 Report
This paper proposes a contactless gesture and sign language behavior sensing method based on ultrasonic signals. A data augmentation method based on GAN is proposed. The authors propose a database and conduct a number of experiments. I give my detailed comments below and hope that my comments can help authors improve their work quality.
1. My biggest concern is the novelty of this paper. The authors only use the common CNN+BiLSTM+CTC framework and propose an open-source database. However, this open-source database has not been publicly released in this paper.
2. Another concern is regarding the unconvincing experimental evaluation. This paper only provides the performance on the proposed database. A comparison with other works published recently and an in-depth analysis of the experimental results should be added, such as [3].
3. In the Related Work Section, the author reviews gesture recognition and sign language recognition in terms of IMU sensors, vision and acoustic. Since there is no clear introduction to single gesture recognition, continuous gesture recognition and sign language gesture recognition, readers may not be able to better understand the motivation of the authors’ work. It is suggested to supplement it. It is recommended to cite and analysis the following references dealing with sign language translation [R1, R2] and sign language generation [R3].
[R1] Hierarchical LSTM for Sign Language Translation, AAAI 2018.
[R2] Graph-Based Multimodal Sequential Embedding for Sign Language Translation, TMM.
[R3] Gloss Semantic-Enhanced Network with Online Back-Translation for Sign Language Production, ACM MM 2022.
4. In Section 3.4, ResNet34, BiLSTM and CTC are the existing method and should not be introduced too much.
5. The content of Section 4.2 is split into too many subsections, resulting in unclear logic. The author should reorganize the writing structure of this part and consider ablation experiments as a separate part.
6. In the Experimentation and evaluation Section, nothing is mentioned about the real implementation details of the recognition network.
7. This paper needs to be re-polished to correct some mistakes and improve the expression.
(1) In Section 3.3, Eq.(5) and Eq.(6) are quoted incorrectly.
(2) The serial number in Algorithm 1 is incomplete and lacks serial number 15.
Reviewer 2 Report
- The techniques used in the paper are not novel enough.
- The networks shown in tests such as Resnet34, VGG16, LSTM are not convincing
- Using only author's own data without reference to other techniques or equivalent data
Reviewer 3 Report
Authors have made significant contribution in the paper. A novel approach is proposed, developed and evaluated by the authors. Results are satisfsctory and impressive.
Reviewer 4 Report
1. The description of the data set is not clear enough. What is the distribution of the training set and the test set?
2.Please add the experimental comparison after adding the CTC algorithm to illustrate the effectiveness of the model improvement.
3. For feature fusion of feature dimension and temporary dimension, please add relevant experiments to explain its function.
